# Chemical Variability and Pharmacological Potential of Propolis as a Source for the Development of New Pharmaceutical Products

**DOI:** 10.3390/molecules27051600

**Published:** 2022-02-28

**Authors:** Piotr Paweł Wieczorek, Nataliia Hudz, Oksana Yezerska, Vladimira Horčinová-Sedláčková, Mariia Shanaida, Oleksii Korytniuk, Iza Jasicka-Misiak

**Affiliations:** 1Department of Analytical Chemistry, University of Opole, 45-052 Opole, Poland; 2Department of Drug Technology and Biopharmacy, Danylo Halytsky Lviv National Medical University, 79010 Lviv, Ukraine; natali_gudz@ukr.net (N.H.); o.yezerska@gmail.com (O.Y.); 3Department of Pharmacy and Ecological Chemistry, University of Opole, 45-052 Opole, Poland; izabela.jasicka@uni.opole.pl; 4Institute of Plant and Environmental Sciences, Slovak University of Agriculture in Nitra, 94976 Nitra, Slovakia; vladimira.sedlackova@uniag.sk; 5Department of Pharmacognosy and Medical Botany, I. Horbachevsky Ternopil National Medical University, 46001 Ternopil, Ukraine; shanayda@tdmu.edu.ua; 6Department of Occupational and Facial Surgery and Dentistry, Ukrainian Military Medical Academy, 01015 Kyiv, Ukraine; koral9999@gmail.com

**Keywords:** propolis, botanical origin, chemical composition, biological properties, health-promoting products

## Abstract

This review aims to analyze propolis as a potential raw material for the development and manufacture of new health-promoting products. Many scientific publications were retrieved from the Scopus, PubMed, and Google Scholar databases via searching the word "propolis". The different extraction procedures, key biologically active compounds, biological properties, and therapeutic potential of propolis were analyzed. It was concluded that propolis possesses a variety of biological properties because of a very complex chemical composition that mainly depends on the plant species visited by bees and species of bees. Numerous studies found versatile pharmacological activities of propolis: antimicrobial, antifungal, antiviral, antioxidant, anticancer, anti-inflammatory, immunomodulatory, etc. In this review, the composition and biological activities of propolis are presented from a point of view of the origin and standardization of propolis for the purpose of the development of new pharmaceutical products on its base. It was revealed that some types of propolis, especially European propolis, contain flavonoids and phenolic acids, which could be markers for the standardization and quality evaluation of propolis and its preparations. One more focus of this paper was the overview of microorganisms’ sensitivity to propolis for further development of antimicrobial and antioxidant products for the treatment of various infectious diseases with an emphasis on the illnesses of the oral cavity. It was established that the antimicrobial activity of different types of propolis is quite significant, especially to Gram-negative bacteria and lipophilic viruses. The present study could be also of interest to the pharmaceutical industry as a review for the appropriate design of standardized propolis preparations such as mouthwashes, toothpastes, oral drops, sprays, creams, ointments, suppositories, tablets, and capsules, etc. Moreover, propolis could be regarded as a source for the isolation of biologically active substances. Furthermore, this review can facilitate partially overcoming the problem of the standardization of propolis preparations, which is a principal obstacle to the broader use of propolis in the pharmaceutical industry. Finally, this study could be of interest in the area of the food industry for the development of nutritionally well-balanced products. The results of this review indicate that propolis deserves to be better studied for its promising therapeutic effects from the point of view of the connection of its chemical composition with the locality of its collection, vegetation, appropriate extraction methods, and standardization.

## 1. Introduction

Beekeeping products have attracted great attention in the scientific community because of their health-promoting properties. Among these products are honey, bee pollen, beebread, beeswax, and propolis [1,2,3,4,5,6,7,8]. Honey possesses well-known health properties and is widely used as a food product [9]. Bee pollen is a valuable source of protein collected by bees from nature for the development of offspring and bee colonies [7]. Beeswax, which is produced by glands located in the bee abdomen, is a valuable component used in cosmetology and pharmacy as a thickener of cosmetic and dosage forms [10].

Propolis is one of the most widely used beekeeping products [1]. The term “propolis” has Greek origin and means defense for (“pro”) the community (“polis”), which refers to a beehive [6,11]. As a resinous substance, propolis is prepared by honey worker-bees (*Apis mellifera* L.) and other bees species for many different purposes. *Apis mellifera* (Western honeybee or European honeybee) is a species of the bee universally managed by beekeepers. *Apis mellifera* has several subspecies or regional varieties, such as the Italian bee (*Apis mellifera ligustica*), European dark bee (*Apis mellifera mellifera*), Carniolan honey bee (*Apis mellifera carnica*), *Apis mellifera caucasica* (F_1_ hybrid), *Apis mellifera* var *ligustica*, etc. [1,12,13,14,15].

Propolis is principally used by bees to seal cracks, smooth walls, avoid the entrance of intruders into hives, and keep moisture and temperature stable in a hive [12,14]. It is collected by bees from the resins and excretions of flowers, leaves, buds, shoots, stems, and fruits. It is obtained after mixing with bees’ saliva [1,14,15]. Propolis is mainly composed of plant resins (60%), pollens, and wax (up to 30%). Among the other organic components are polyphenols and essential oils [16,17]. Flavonoids, phenolic acids, and terpenoids are regarded as the main biologically active substances of propolis [14,15,18,19]. Okińczyc et al. [19] supposed that the therapeutic effects of propolis are related mainly to the amount of polyphenols and volatile components. Fatty acids, fatty acid esters, amino acids, enzymes, sugars, vitamins (B_1_, B_2_, B_3_, and B_6_), Fe, and Zn are components of propolis as well [20,21,22].

The latest studies on the biological activity of propolis point to a multitude of its activities. It is worth mentioning that crude propolis should be subjected to toxicological testing before applying these samples for the preparation of dietary supplements and medicinal products [22]. Propolis is a nontoxic product. Its safe concentration for humans is approximately 1.4 mg/kg per day or 70 mg/day [23]. In recent years, beekeeping products have been widely applied in traditional medicine. Propolis extracts have positive repercussions in the treatment of several diseases [6]. The list of biological activities of propolis is very wide. Among them are antiseptic, antibacterial, antimycotic, antiviral, antiprotozoal, antioxidative, spasmolytic, choleric, astringent, anti-inflammatory, anesthetic, antitumor, immunostimulating, cytostatic, hepatoprotective, and other activities [4,14,24,25,26].

Many researchers have studied the chemical composition of propolis and established different correlations between its composition and plants used for producing propolis [19]. Other scientists have searched for methods of treatment of diseases caused by drug-resistant microorganisms and oxidative stress [14,27,28]. Among the aspects of the standardization of propolis products are extraction processes. Different solvents and types of maceration are widely used in the extraction of biologically active substances of propolis. In this review, the composition and biological activities of propolis are presented from a point of view of the development of new medicinal products on the basis of propolis, including its standardization. Additionally, the therapeutic strategies of propolis are also discussed. Finally, as the antimicrobial activity of different types of propolis is noticeable, one more focus of this paper is an overview of the chemical composition and microorganism sensitivity to propolis for the purpose of the elaboration of antimicrobial and antioxidant pharmaceutical products for the treatment of various infectious diseases.

### 1.1. Historical Aspects of Propolis

The term “propolis” means “suburb/bee glue” or “defense of the city”, derived from Hellenistic ancient Greece depending on the interpretation [13]. The use of propolis dates back to at least 300 BC when it was used by Egyptians, Persians, Greeks, and Romans [29,30]. It was employed for the mummification by Egyptians and as a topical cream for the treatment of wounds, cuts, ulcers, and other dermatological problems. It remained mostly as an alternative herbal medicine, mainly in Eastern Europe (Russia), where it later got the name “Russian penicillin” [13]. The growing popularity of the usage of propolis was rediscovered again in the Renaissance by ancient teachings and medicine. The first scientific research of propolis started in the 19th century with its distillation, and the first major chemical research was conducted at the beginning of the 20th century with its fractionation [13,29]. 

The first isolated constituents from propolis were vanillin, cinnamic acid, and cinnamyl alcohol [13]. Propolis has been intensely studied for the last decades, and up to the year 2000, at least 300 compounds have been identified in propolis [16,31,32,33], and 112 flavonoids were identified in various types of propolis from 2000 to 2012 [16]. Between 2013 and 2018, at least 305 compounds were isolated from propolis for the first time, including the isolation of alkaloids [31]. Altogether, up to the year 2018, more than 850 compounds have been reported from propolis [32,33]. In available publications, we have not found the information on how many compounds have been found from 2018 until now.

### 1.2. Origin of Propolis

The bioactive components of propolis originate from plant exudates, which contain propolis precursors [19,25,30]. Thus, the plant species used by bees for the production of propolis play a very important role in the chemical composition and biological properties of the final product (propolis) [19,30]. As a rule, bees show preference for certain species of plants for foraging, depending on a region. Therefore, the chemical composition of propolis is not completely random, and it can be divided into several chemical types [19]. To date, numerous types of propolis have been described. Among them are European propolis of the poplar type, Mediterranean propolis, Brazilian green propolis, Brazilian red propolis, Canadian propolis, Venezuelan propolis, Chinese propolis, Argentinian propolis, Turkish propolis, Algerian propolis, Egyptian propolis, Mexican propolis, Greek propolis, etc. [14,15,21,34,35,36,37,38,39]. The characteristics of each type of propolis depend on plant source, edaphoclimatic conditions, season, bee species, etc. [1,14,37,40,41]. 

Nowadays there exists a conception that the type of propolis is commonly related to the defined plant as the main source for obtaining propolis of an appropriate type. This plant contains inherent biologically active substances that could be considered as markers of propolis and, thus, its pharmaceutical preparations. The most common propolis collected from Europe, North America, non-tropical regions of Asia, New Zealand, and even Africa has poplar chemical characteristic such as high flavonoid content, including the flavone content, and low phenol and ester concentrations that could be explained by the tendency of gathering bud exudates of poplar trees by *Apis mellifera* [1,19]. According to Bankova et al., 2014 and Okińczyc, 2018, the basic plant sources of propolis in the temperate zone are the bud exudates of trees of the genus *Populus*, chiefly the black poplar (*P. nigra)*, and more rarely aspen *(**P. tremula* L.). These authors named this propolis European propolis. Such typical poplar phenolics such as flavones, flavanones, and esters of substituted cinnamic acids are the principal components of European propolis [19,31,37,40]. Sometimes, a white birch (*Betula verrucosa*) is also the source of propolis [19,37]. However, there are also known mixed types of propolis: aspen–poplar, aspen–birch–poplar, etc. [19]. Some authors divide propolis from Europe into such subtypes: European brown poplar propolis, Mediterranean propolis, Irish propolis, Czech propolis, German propolis, Ukrainian propolis, Polish propolis, Greek propolis, Croatian Mediterranean propolis, and Croatian continental propolis, etc., according to the country or even its part in which propolis was collected [4,19,26,37,42,43,44]. However, it is worth noting that El-Guendouz et al., under the term Mediterranean propolis, understand propolis from all the countries surrounded by the Mediterranean Sea (Southern Europe Coast, Levantine Coast, and Northern African Coast) [45]. For that reason, the classification according to the country is conditional to a certain extent, especially in the case of propolis collected in the countries which are surrounded by the Mediterranean Sea as, for instance, Croatian propolis can be European or Mediterranean.

The main source of Cuban, Brazilian Amazonian, and Venezuelan propolis is *Clusia* spp., with prevaluated polyprenylated benzophenones [46,47]; Brazilian green propolis—*Baccharis dracunculifolia*, with prenylated phenylpropanoids, caffeoylquinic acids, and diterpenes as major compounds [48,49]; Brazilian red propolis—*Dalbergia ecastophyllum* with isoflavonoids, neoflavonoids, pterocarpans, and lignans as predominant components [14,34,50,51,52]; Brazilian brown propolis—*Hyptis divaricata*, Brazilian poplar type—*Populus alba*, and Pacific propolis—*Macaranga* spp., which contains mainly *C*-prenyl-flavanones [53,54]; and Sicilian, Greece, and Maltese propolis (Mediterranean propolis)—conifers (*Cupressaceae*), with diterpenes (mainly acids of labdane type) [37,45,55,56,57,58]. Prenylflavonoids (solophenol A, bonannione A, sophoraflavanone A, and (2*S*)-5,7-dihydroxy-4’-methoxy-8-prenylflavanone) were identified in the propolis collected from Malaita Island in the Solomon Islands [59]. The main source of Russian propolis is catkins of *Betulla verrucosa*, with major compounds of flavones and flavonols, which are different from the poplar type [29,37,58,60]. Nonetheless, such a correlation between the type of propolis and its main botanical source is conditional to some extent. For example, samples of Brazilian green propolis from different regions of Brasilia can have quite different compositions, even despite a common plant origin (*baccharis dracunculifolia*). such prenylated phenylpropanoids as artepillin c (3,5-diprenyl-4-hydroxycinnamic acid) and 3-prenylcinnamic acid allyl ester are considered to be the markers of brazilian green propolis, the most exported one [57,58]. in addition, dicaffeoylquinic acids could be considered as markers of brazilian green propolis as well [57]. isoflavones can be considered as markers of the red brazilian propolis [34]. honey bees produce red propolis in some continents: africa (nigeria), asia (saudi arabia), north america (cuba, and mexico), and south america (brazil, and venezuela) [14,34,36,57]. 

Propolis of african stingless bees (*meliponula ferruginea*) is rich in biologically active substances as well. The content of sugars, fatty acids, and diterpenes is 9.1–15.6%, 2–4.6%, and up to 60.6%, respectively [41].The uncommon propolis collected by stingless bees of the *Meliponini* tribe (*Scaptotrigona postica)* is a mixture of resin, wax, and soil known as geopropolis [61]. The Baha sample (Saudi Arabia) contains a large amount of aromatic acids, alcohol, and phenol aldehydes. In addition to these components, some other compounds (aliphatic acids, sugar derivatives, steroid derivatives, and flavone derivatives) were also present [30].

The above-mentioned information points out that the chemical composition of extracts prepared from propolis will mainly depend on the plant sources used by bees for forming propolis, as well as species of bees [19,30,37,58]. Some authors connect the chemical composition of propolis with the vegetation of the locality [19,26,34,37,58]. Nevertheless, it can be supposed that the vegetation is constant to a certain extent for the particular area, and actually, the locality can be regarded as the principal factor that mainly determines the chemical composition of propolis. Thus, plant sources and species of bees could be regarded as principal factors that have an influence on the chemical composition of propolis and its preparations. Moreover, there are some natural complementary factors affecting the chemical composition of propolis. Among them are climate, season, and year [14,30]. Additionally, there are some complementary technological factors that can have an impact on the chemical composition of propolis preparations. For instance, the chemical composition of extracts also can depend on the extraction solvent, ratio of a solvent to propolis, and extraction procedures (maceration, remaceration, temperature of extraction, sonication, etc.) that are very important for pharmaceutical technology [4,5,21,42,43]. Therefore, we can suppose that pharmaceutical products prepared from propolis of diverse origins may have dissimilar polyphenolic compositions and, therefore, different biological activity and quality. For this reason, when developing pharmaceutical products, it is necessary to take into consideration the locality of obtaining propolis at least on the level of a country, to use standardized methods of their preparation, and to elaborate analytical procedures for the evaluation of the total phenolic and flavonoid contents and content of special markers characteristic for the special type of propolis related to the country of the collection of propolis. 

### 1.3. Extraction Procedures

Since the activity of propolis preparations as other herbal preparations in particular depends on the extraction of active substances, we provide a brief analysis of extraction techniques for propolis with the emphasis on solvents, their concentration, and extraction regime that are important for their application in the pharmaceutical research with the purpose of the development of standardized products of propolis.

As a rule, propolis extracts are prepared using continuous soaking in various solvents: water, acetone, ethanol of different concentrations, a mixture of methanol and dichloromethane, etc. [35,43,62,63,64,65,66]. Ethanol is mainly used for extraction, especially in the food and pharmaceutical industry [37,62] or for analytical and/or microbiological studies [4,5,19,37,43,63,65,66]. There are other techniques, including ultrasonic and microwave extraction [1,19,64]. In general, the nature of the solvent, temperature, and time of extraction, sonication, composition, and physical characteristics of propolis samples have significant influences on the yield of extraction and quality of final preparations [21,42,43,63,66]. In addition, 70% ethanol is widely employed in the studies of propolis [4,19,43,47,63,66], which could be explained by some facts. Firstly, ethanol of this concentration extracts flavonoids better compared to water and 96% ethanol [63,66]. Secondly, 50% and 70% ethanol are widely used in pharmaceutical technology [4,5].

Cottica et al. studied the influence of extraction type and solvent on the total phenolic content (TPC) and total flavonoid content (TFC), antioxidant activity, and sensory analysis of the commercial Canadian propolis. It was shown that the type of solvent and the number of extractions had an influence on the yield of extraction, TPC, and TFC. They established that polyphenols and flavonoids were preferentially extracted by ethanol for the two modes (ethanol plus water and oppositely), causing an improved antioxidant capacity. These authors also revealed that double extraction (remaceration) is more effective than single extraction [35]. 

Woźniak et al. showed that the solvent used for extraction (acetone 100%, ethanol 96%, and 70%) affected the concentration of flavonoids, phenolic acids and the antioxidant, antifungal, and cytoprotective effects of the extracts obtained from Polish propolis. The flavonoid concentrations in the tested extracts varied depending on the solvent used. The propolis acetone extract was characterized by the highest TFC (95.25 mg/g of propolis extract). The TFC in the extracts in which 70% and 96% ethanol were the solvents was equal to 93.13 and 76.13 mg/g of propolis extract, respectively. The galangin concentration was the highest in the acetone extract, whereas chrysin and pinocembrin concentrations were the highest in the extract in which 70% ethanol was the solvent, and the highest concentration of kaempferol was detected in the extract in which 96% ethanol was the solvent. The highest concentration in all the propolis extracts among phenolic acids was observed for coumaric acid. The sum of phenolic acids was 17.75, 17.88, and 18.49 mg/g of the propolis extract if acetone, 70% ethanol, and 96% ethanol, respectively, were the solvents. All the propolis extracts exhibited high and statistically similar free-radical-scavenging activity, Fe^3+^-reducing power, and ferrous ion (Fe^2+^)-chelating activity. The propolis extract obtained using 70% ethanol for the extraction showed higher activity against fungi in comparison to the extract for which 96% ethanol was used as a solvent [63]. Therefore, it can be supposed that 70% ethanol is the most suitable for the preparation of the propolis extracts.

The propolis samples of Ethiopian origin were extracted with a mixture of dichloromethane and methanol for the purpose of the extraction of polar and nonpolar components. The results demonstrated that the total extract yields were in the range of 27.2% to 64.2%. The principal components were triterpenoids (85.5 ± 15.0% of the total extracts, mainly α- and β-amyrins and amyryl acetates), *n*-alkanes (5.8 ± 7.5%), *n*-alkenes (6.2 ± 7.0%,), methyl *n*-alkanoates (0.4 ± 0.2%), and long-chain wax esters (0.3 to 2.1%). Moreover, phenolics were not identified in these samples [64]. The yields of hexane extracts ranged from 16.19% to 71.16% for the samples of propolis from Algeria. The ratio was 1 part propolis to 30 parts solvent. These propolis extracts were intended for the identification and semi-quantification of fatty acids [21]. The yields of the ethanolic and aqueous extracts were 39.53% and 28.05%, respectively, for the commercial Canadian propolis at a 1:10 ratio of propolis to solvent [35]. Therefore, the yield mainly depends on the sample of propolis and solvent, and it is impossible to compare results if different solvents and ratios are used.

Al-Ani et al. compared ethanolic and aqueous extracts of propolis from Germany in terms of their antimicrobial activity, TPC, and TFC. The aqueous propolis extract was prepared in the following way: 10 g of very fine dried powder was dissolved in 20 mL of sterile water. The maceration was kept at a temperature of 60 °C for 7 h. Then was a separation by centrifugation, filtration by filter paper, and evaporation with low pressure to dispose of excess water. In a similar way, the ethanolic extract was prepared: 10 g of the propolis mixed in 100 mL of 70% ethanol and shaken at a temperature of 37 °C for 24 h. Then was separation by centrifuge, filtration by filter paper, and evaporation with low pressure to dispose of excess ethanol. It was established that the ethanolic extract contained a significantly higher TPC and TFC in spite of a significantly larger mass of propolis at the preparation of the aqueous extract (46 mg caffeic acid equivalents per gram of the ethanolic extract versus 10 mg per gram of aqueous extract, and 1.9 mg of quercetin equivalents per gram of ethanolic extract versus 0.1 mg of quercetin equivalents per gram of aqueous extract). The larger yield of phenolic acids and flavonoids in the ethanolic extract could be explained by the better extraction by ethanol compared to water. Despite a significant difference in the TPC and TFC, the ethanolic extracts of propolis of Germanic, Irish and Czech origin and aqueous extract of Germanic origin showed moderate effects against human respiratory bacterial pathogens, including positive β-lactamase production *Haemophilus influenzae* (MIC between 0.6 mg/mL to 2.5 mg/mL) and *Streptococcus pneumoniae* (MIC between 0.08 mg/mL to 0.6 mg/mL). In general, all the ethanolic extracts exhibited remarkable bactericidal activity against Gram-positive microorganisms with MIC between 0.08 mg/mL and 5 mg/mL, while only some ethanol extracts of propolis showed moderate efficacy against Gram-negative microorganisms, with MIC between 0.6 mg/mL and 5 mg/mL. *Pseudomonas aeruginosa* was highly resistant towards propolis [43]. 

For sublingual administration by volunteers, Lisbona-González et al. used an extract prepared as follows: 20 g of unrefined propolis was crushed and dissolved in 100 mL of 66% ethanol. The maceration lasted at room temperature for 28 days [65]. For the study of antiviral activity, 1 kg of the sample of geopropolis was extracted by maceration in 1 L of ethanol for 3 months. Then this alcoholic extract was filtered and concentrated and then stored in a freezer [61]. However, such long periods (1–3 months) of extraction are not suitable for the pharmaceutical industry. The ethanol concentration has a significant impact on the extraction of phenolics [43,66]. For example, for Romanian propolis, as a poplar type propolis, the phenolics profile of 25% ethanol extracts is similar to the aqueous one, with the exception of the quercetin content. The quercetin content could explain the increase in the antimicrobial activity concerning the Gram-positive bacterium *Bacillus subtilis* (MIC < 805 μg/mL), along with the demonstrated significant contribution to the antioxidant activity of the extracts. The 50% ethanolic extracts contain, in addition to phenolic acids, large quantities of chrysin and galangin. Galangin and chrysin are in small amounts in the higher water-containing solvents. It seems that the high flavonoid levels are related to the better antibacterial activity towards *Bacillus subtilis*. In general, the results of the experimental studies demonstrated that 50% ethanolic extracts have a rich polyphenolics profile and, therefore, a good antioxidant capacity [66].

Galeotti et al., 2018 studied the special propolis extract dissolved in various solvents, obtaining different final products such as hydroalcoholic, glycolic, and glyceric solutions, oily products, as well as a product in the form of a powder. The extracts had quite similar polyphenol quantitative and qualitative compositions. However, the powder was richer in phenolic acids (caffeic, coumaric, ferulic, and isoferulic) than liquid preparations, ~10% versus 0.5% because of the specific and owner production procedure. All of the five products contained pinobanksin, chrysin, pinocembrin, galangin, pinobanksin-3-*O*-acetate, pinobanksin-3-*O*-butyrate, and various other pinobanksin derivatives. Overall, the sum of flavones and flavonols was 36.4%, 25.4%, 25%, and 19.9%, respectively, in the glyceric, glycolic, hydroalcogolic, and oily extracts, while the sum of flavanones and dihydroflavonols was 33.4%, 37.4%, 39.5% and 41.3%, respectively. ESIT12 is a micronized sample with a TPC of a minimum of 12%, which is used for the preparation of tablets and capsules [42].

The results of reviewing the publications connected with propolis confirmed that water and ethanol are usually used as the main solvents for propolis in the studies related to antimicrobial activity and in studies on people [35,43]. However, considering the antimicrobial activity of ethanol compared to water, alcoholic extracts using 50–70% ethanol are preferable for the pharmaceutical industry because of their better microbiological and chemical stability. Moreover, ethanol extracts contain more phenolics that have biological activity. Therefore, tablets, capsules, and oral sprays based of ethanolic propolis extracts can be regarded as complementary preparations for the treatments of infections of the oral cavity and upper respiratory organs, which are especially induced by Gram-positive bacteria.

## 2. Biologically Active Compounds Presented in Propolis

### 2.1. Terpenoids

Terpenes have antimicrobial, anti-inflammatory, and anticancer potential [23,25]. Therefore, propolis preparations could be regarded as preparations of at least complementary therapy of some diseases.

#### 2.1.1. Mono- and Sesquiterpenoids (Volatiles)

Volatiles were found in propolis in low concentrations (up to 1% mainly), but their aroma and prominent biological properties make them important for the characterization of propolis [40]. As it is known, essential oils are complex mixtures of volatile constituents, mainly terpenes and phenylpropanoids. The phytochemical analysis of two essential oils hydrodistillated from Algerian propolis showed cedrol (17.0%) dominating in the samples from Oum El Bouaghi, while α-pinene (56.1%) dominated in the samples from Batna [67]. β-Caryophyllene and nerolidol were the predominant compounds of Argentinian green propolis [68]. α-Pinene and β-pinene were the principal compounds of different samples of the propolis from Brazil [18]. Limonene was identified as the chemical marker of Venezuelan propolis [68]. Among the volatile components of the Polish propolis were benzyl alcohol, *cis*-β-caryophyllene, caryophyllene oxide, *trans*-nerolidol, benzyl benzoate, and salicyl benzoate [19].

The essential oil composition of propolis from twenty-five Chinese locations in total contained 406 compounds, and principal component analysis found a significant correlation between the composition and origin of propolis samples. Among the major components were cedrol, γ-eudesmol, phenethyl alcohol, benzyl alcohol, 2,3,4-dimethoxystyrene, methoxy-4-vinylphenol, and guaiol [69].

It should be mentioned that α- and β-pinene demonstrated significant anticonvulsant effects in the model of pentylenetetrazole-induced convulsions in the animals [70]. Cedrol demonstrated prominent anti-inflammatory and analgesic effects in mice [71]. The application of D-limonene is mainly due to its high-quality fragrance property [72]. This monocyclic monoterpene possesses antioxidant, antidiabetic, anti-inflammatory, anticancer, cardioprotective, gastroprotective, and immune-modulatory effects. Eudesmol has the potential for anti-tumor activities by inhibiting angiogenesis [73]. The recent advances in the areas of anticancer and antimicrobial activity of mono- and diterpenes were demonstrated by Greay and Hammer [74].

Moreover, some samples with the highest percentage of diterpenes (80.4%, 73.3%, and 81.9%, respectively) from Western Crete and Greece demonstrated the highest antimicrobial activity against all the tested microorganisms, especially against Gram-positive bacteria (*Staphylococcus aureus*, *Streptococcus epidermidis*, and *Streptococcus mutans*). It is worth mentioning that these samples had a very small content of flavonoids or even a zero level [37]. For that reason, standardized propolis preparations in the form of tablets, capsules, sprays, or suppositories could be administered for the complementary treatment of inflammatory diseases, cancer, and infectious diseases. 

The structures of some mono- and sesquiterpenoids identified in propolis are provided on Figure 1.

#### 2.1.2. Diterpenes

Propolis is one of the significant sources of bioactive diterpenes [30,37,75]. Diterpenes are well known for their bioactive effects such as anticancer, antibacterial, and anti-inflammatory. Active clerodane diterpenes possessing anticancer activity were isolated from Brazilian propolis. The derivatives of clerodane-type diterpenes, as well as labdane diterpenes, act as antimalarial and anti-inflammatory drugs [75].

Pollen analysis of the studied Greek propolis samples showed that approximately 90% of the samples rich in diterpenes originated from Coniferae trees, especially from *Pinus* sp. Therefore, the propolis samples from Greece had features that distinguished them from the typical European propolis for the high content of diterpenes and relatively low quantity of phenolic acid esters and flavonoids. Such features give the possibility to consider propolis from Greece as a new type of propolis (Mediterranean propolis) consisting mainly of diterpenes and produced on the base of Conifer trees among *Cupressaceae* and *Pinaceae*, which are widely spread in the Mediterranean area [37]. Among diterpenes of the Baha propolis from Saudi Arabia were identified cembrene (C_20_H_32_) and totarol (C_20_H_30_O) [30].

The structures of some diterpenoids are provided on Figure 2.

#### 2.1.3. Triterpenoids

The major compounds of the extracts of propolis of Ethiopian origin were triterpenoids (85.5 ± 15.0% of the total extracts, chiefly α-amyrin, and α-amyryl- and β-amyryl acetates), *n*-alkanes (5.8 ± 7.5%), *n*-alkenes (6.2 ± 7.0%,), methyl *n*-alkanoates (0.4 ± 0.2%), and long-chain wax esters (0.3% to 2.1%). In addition, α-amyrin dominated (83.79% and 63.11%) in the two samples of Ethiopian origin (Enemore and Holleta areas) and α-amyryl- and β-amyryl acetates in other two samples (29.72–53.79%) (Bako and Gedo areas). Rushdi et al. suppose that triterpenoids are dominant components of propolis from tropical and semi-tropical regions. Their opinion was based largely on the evidence that the highest triterpenoid concentrations were revealed in the propolis from the Bako (97.6%) and Gedo (93.7%) zones, where the principal vegetation is dominated by *Acacia* species, *Euphorbiaceae* species (*Croton macrostachys*), and *Boraginaceae* species (*Cordia africana*) [64]. Lupenone, α-amyrin, and β-amyrin were identified in the hexane fraction of propolis of Nkambe (northwest Region, Cameroon) [76]. The structures of amyrins are provided on Figure 3.

### 2.2. Phenolic Compounds

Scientists found that the pharmacological activities of propolis are due to the presence of phenolic acids, their esters, and flavonoids, which have the highest antiradical activity [19,42,43,63,77]. The significant antioxidant and antibacterial activities of propolis samples gathered in Palestine and Morocco correlated with their high total phenolic and flavonoid contents [78]. Different approaches to the standardization of propolis were applied by the Ukrainian researchers [4,79]. Chromatographic investigations showed the high diversity of phenolic molecules, which are regarded to be the main bioactive compounds of propolis and were proposed as the standardization markers [79]. 

Flavonoids and phenolic acids are the most representative biologically active compounds in propolis, at least European propolis [12,19,80]. In general, TPC and TFC are the parameters of great importance as they are related to the biological activity of a natural product, especially antimicrobial potential [3,4,5,12,44]. However, the qualitative and quantitative composition of polyphenols depends on the season, the locality of hives related to regional flora, the material of a beehive, type, and solvent of extraction [12,43]. For example, MALDI spectra revealed that among the principal components of the Mongolian propolis were aromatic acids (cinnamic acid and *p*-coumaric acid), dihydrochalcones (2,4,6-trihydroxydihydrochalcone), fatty acids (stearic acid, palmitic acid), and esters (benzylmethoxybenzoate) [81]. 

### 2.3. Flavonoids

Propolis of different types is rich in polyphenolic compounds, chiefly flavonoids, cinnamic acids, and their esters [19,30,38,39,42,43,44,82]. In general, phenolic compounds, including flavonoids, are associated with the antimicrobial and anti-inflammatory activity of propolis extracts [4,14,38,44,66]. Flavonoids are among the principal polyphenols in propolis, which is considered as a main criterion to evaluate the quality of propolis [1,4,16,30,36,44].

Tectochrysin (techtochrysin) was the most abundant flavonoid aglycone, with content of up to 16.07 mg per mL of the extract of Croatian propolis. One of the tested samples showed the highest content of some flavonoids, namely, tectochrysin (16.07 mg/mL), galangin (8.71 mg/mL), pinocembrin (6.39 mg/mL), chrysin (8.02 mg/mL), apigenin (1.23 mg/mL), and kaempferol (0.67 mg/mL). The content of ferulic and *p*-coumaric acid was 1.37 and 1.03 mg/mL, respectively. In general, such flavones as apigenin and chrysin were identified in 20 samples out of 24, and flavonols such as galangin—in 18 samples out of 24 [44]. Chrysin, galangin, and pinocembrin were identified in the ethanolic extracts of propolis of Irish and Czech origin as well [43].

Poplar propolis is rich in flavones, flavanones, and phenolic esters [60]. All the groups of flavonoids were revealed in propolis (flavones, flavonols, flavanones, flavanonols, chalcones, dihydrochalcones, isoflavones, isodihydroflavones, flavans, isoflavans, neoflavonoids, and flavonoid glycosides) [1,12,34,36,38,44,82,83]. Flavanones identified included pinocembrin (2–4%) and naringenin [36,38,83]; flavanonols included pinobanksin [36] and its derivatives (pinobanksin-3-*O*-acetate, pinobanksin-3-*O*-butyrate) [42]; flavanols comprised catechin (5.8% in Mexican propolis) [38]; flavones included chrysin (2–4%), tectochrysin [44,82,83], chrysin (5.3% in propolis extract) [36], 24.4% in Mexican propolis [38], apigenin, luteolin, vitexin [14,42,44,82], and luteolin-5-methyl ether [84]; flavonols embraced rutin, quercetin, kaempferol [38,42,44,82], galangin (1–2%) [36]; and chalcones included 2′,3′,4′-trimethoxychalcone, 2′-hydroxy-3′,4′-dimethoxychalcone, and 2′,4′-dihydroxy-3′-methoxychalcone [15,81]. In addition, among flavonoids of the extract of Czech propolis were kaempferol (101 mg/L ± 45 mg/L), apigenin 73 mg/L ± 8 mg/L, and chrysin 36 mg/L ± 5 mg/L. Moreover, pinocembrin, galangin, naringenin, luteolin, genistein, and quercetin were not detected [85]. Furthermore, in one more study it was revealed that propolis collected in the Van lakeshore (Turkey) contained daidzein, rutin, and luteolin in a concentration of 56.5, 34.7 and 3.60 mg per 100 g of the propolis sample, respectively [12].

In Nepalese propolis, such flavonoids as cearoin, chrysin, 3′,4′-dihydroxy-4-methoxydalbergione, isoliquiritigenin, 7-hydroxyflavanone, 4-methoxydalbergion, plathymenin, and (+)-medicarpin were identified. The results of biological studies showed that 3′,4′-dihydroxy-4-methoxydalbergione,4-methoxydalbergion, cearoin, and chrysin induced the anti-inflammatory activity of Nepalese propolis [82].

According to the results of the chromatographic analysis, all the samples of propolis from Greece were divided into two groups: (1) rich in diterpenes, and (2) rich in flavonoids with a low content of diterpenes [37]. The flavonoids were the same as in the well-known poplar type, European propolis. Chrysin, galangin, pinocembrin, pinobanksin, and pinobanksin-3-*O*-acetate were in large quantities. These components came from bud exudate of the genus *Populus*, chiefly *Populus nigra*. The samples of the “flavonoids-rich” group were from the regions where *Populus nigra* grows. The low flavonoid content in the samples from the “diterpenic” group could be explained by the small spreading of poplar in the respective places of collection [37]. Thus, Greek propolis can be European or Mediterranean depending on the place of its collection, related in turn to vegetation. This confirms that a part of the country can have an influence on the chemical composition of propolis. 

The presence of quercetin and kaempferol in propolis can be explained by fact that more than 50% of plants contain these aglycones [3]. In general, flavonoids represent more than 60% of the composition of the Brazilian red propolis [14]. The principal flavonoids identified in geopropolis were flavones di-*C*-glycosides, and among them were apigenin-6,8-di-*C*-malonyl glucoside dihexoside isomer, apigenin-di-*C*-malonyl trihexoside isomer, vicenin-2 (apigenin 6,8-diglucoside), chrysin-8-C-rhamnoside-7-*O*-rhamnoside, luteolin-8-*C*-caffeoyl rhamnoside, etc. The sugar residues of flavonoids glycosides are chiefly presented by hexoses (galactose and glucose), 6-deoxyhexoses (furanose and rhamnose), pentoses (arabinose and xylose), and uronic acids (glucuronic acid and galacturonic acid) [61].

Gardana et al. provide such a correlation between the TFC and characteristic of propolis quality: less than 11%, 11–14%, 14–17%, and more than 17%, and, respectively, low, acceptable, good, and high quality [36]. However, it seems that this classification of propolis quality is not convenient, as liquid chromatography coupled to diode array detection and mass spectrometry (LC–DAD–MS) analysis and a huge number of standards are needed in a laboratory. Moreover, such analyses are expensive and time-consuming for routine standardization. Furthermore, flavonoids are regarded as minor components of Brazilian propolis [57]. Finally, flavonoids were absent in propolis from Ethiopia [64]. From the point of view of Rebiai et al., 2021, the determination of total polyphenols by the spectrophotometric method and HPLC might be considered important analytical methods for the evaluation of propolis content [86]. To our mind, TPC and TFC could be important indexes of the quality of propolis with a high content of flavonoids. 

The structures of some flavonoids are provided in Figure 4.

### 2.4. Phenolic Acids

Among phenolic acids in the Slovak propolis were caffeic acid, *p*-coumaric acid, ferulic acid, and cinnamic acid [82]. A similar composition of phenolic acids was in the extract of the Czech propolis, which was confirmed in neighbors of these two countries and its common European origin. Among phenolic acids were determined caffeic acid (65 mg/L ± 11 mg/L), *p*-coumaric acid (231 mg/L ± 10 mg/L), *t*-ferulic acid (514 mg/L ± 15 mg/L), and *t*-cinnamic acid (29 mg/L ± 1 mg/L) [85]. One more study proves that propolis is rich in caffeic, coumaric, ferulic, and isoferulic acids (9.7% in the propolis extract) [42]. Galeotti et al. named this propolis European brown poplar propolis [42]. 

HPLC analysis of the Brazilian red propolis produced by Africanized *Apis mellifera* and collected in the state of Pernambuco revealed nine phenolic compounds. Among them were flavones and flavonols (apigenin, luteolin, vitexin quercetin, and rutin) and phenolic acids (caffeic acid, chlorogenic acid, ellagic acid, and *p*-coumaric acids) [14]. Such phenolic acids as gallic, caffeic, chlorogenic, and coumaric were detected in Algerian propolis [86]. It seems that these phenolic acids are common for any propolis.

### 2.5. Other Organic Compounds

#### 2.5.1. Alkanes and Alkenes

The concentrations of *n*-alkanes in the propolis samples from Ethiopia were in the range of 0.87% to 16.9% of the total extracts (5.82 ± 7.48%). Moreover, the predominant *n*-alkanes ranged from C_21_ to C_31_. Heptacosane (C_27_H_56_) was present in the largest amounts (0.44–7.56%). In addition, plant wax *n*-alkanes usually have a C_max_ in the range of C_25_–C_31_, which varies depending on the plant species, the season, and locality. Therefore, the odd carbon number preference of the C_21_–C_31_
*n*-alkanes and the Cmax at 27 point towards the supposition that the major sources of these *n*-alkanes are likely from beeswax. The contents of the *n*-alkenes were in the range of 0.85% to 15.92% (6.23 ± 6.96%). The largest concentration (15.92%) was determined in the propolis sample from the Holleta area and the smallest (0.85%) in the samples from Bako. The number of carbon atoms was in the range of C_25_ to C_36_ with a C_max_ of 33. The presence of *n*-alkenes with large concentrations of the odd-numbered homologues and C_max_ of 33 corroborates their origin from insect wax, possibly from the alteration of long-chain *n*-alkanols [64].

#### 2.5.2. Fatty Acids

Fatty acids are components of propolis as well [21,30,76]. Propolis from the six regions of Algeria contained such major acids as *cis*-11-eicosenoic acid (C20:1, *n*-9), *cis*-11,14-eicosadienoic acid (C20:2, *n*-6), *cis*-5,8,11,14,17-eicosapentaenoic acid (C20:5), arachidonic acid (C20:4, *n*-6), linoleic acid (C18:2, c + t *n*-6), palmitoleic acid (C16:1), and *a*-linolenic acid (C18:2, c + t *n*-6) [21]. 

Oleic acid, nonanoic acid (C_9_H_18_O_2_), decanoic acid (C_10_H_20_O_2_), and dodecanoic acid (C_12_H_24_O_2_) were also identified in the Baha sample of propolis from Saudi Arabia [30]. Hexatriacontanoic acid (C_36_H_72_O_2_) was identified in the sample of propolis from Cameroon [76].

Therefore, propolis could be regarded as a source of omega fatty acids and, therefore, active substances of pharmaceutical preparations for the prevention of some diseases. 

The components and appropriate type of propolis are presented in Table 1.

## 3. Main Biological Properties

### 3.1. Antioxidant Activity

As oxidative stress causes many diseases, the antioxidant capacity of beekeeping products is very important and can be considered as a clue factor for the elaboration of medicinal products. Therefore, there is a need to establish standardized methods for evaluating their free-radicals scavenging potential. For such evaluation, the DPPH (1,1-diphenyl-2-picrylhydrazyl) free radical-scavenging assay, cupric ion reducing antioxidant capacity assay, and ferric reducing antioxidant power assay are widely used [63,77]. As is known, polyphenols possess a large antioxidant potential among natural phytochemicals [9].

High concentrations of phenolics are responsible for the antioxidant, antimicrobial, and anti-fungal activities of the extracts of Romanian propolis, which were prepared on the basis of 50% and 70% ethanol. However, the extraction with 70% did not achieve benefits compared to the 50% ethanol extracts [66]. All the extracts of propolis of Polish origin (aqueous, ethanolic on 70% and 96% ethanol) showed high and statistically similar free-radical-scavenging activity, Fe^3+^-reducing power, and ferrous ion (Fe^2+^)-chelating activity. The results indicated that the solvent (water, 70% ethanol and 96% ethanol) used in the extraction process did not have an influence on the DPPH free-radical-scavenging activity, Fe^3+^-reducing power, and ferrous ion (Fe^2+^)-chelating activity of all the extracts. Moreover, all the propolis extracts at the concentration of 0.05 mg/mL considerably protected human erythrocytes under oxidative stress against oxidative hemolysis caused by peroxyl radicals generated from 2,2′-azobis-(2-methylpropionamidine) dihydrochloride [63]. 

The antioxidant activity (IC_50_) of the ethanolic extracts of the Czech, Irish, and Germanic propolis was 27.72, 26.45, and 32.53 µg/mL, respectively, while under the same conditions the value of IC_50_ was 3.21 µg/mL and 36.4 µg/mL for ascorbic acid and aqueous extract of Germanic propolis, respectively. The Irish propolis showed the highest value of TFC (2.86 mg quercetin equivalents per 1 g). The results of this study revealed that ethanolic extracts had a slightly higher antioxidant activity that could be explained by higher concentrations of extracted flavonoids [43,66]. The antioxidant activity was similar for the oily, glycolic, glyceric, and ethanolic of propolis, showing that preparations exerted antioxidant capacity and that this activity is connected to the TPC of propolis products [42].

One more study tested the influence of Turkish propolis and its nano form on the expression levels of anti-apoptotic and pro-apoptotic proteins in the testes of the rats treated with cisplatin, sperm quality, reproductive organs, and antioxidant status. Cisplatin is a chemotherapeutic active substance related to oxidative stress and apoptosis. Cisplatin may have adverse effects on the reproductive system. As a result of this study, it was revealed that propolis and its nanoform (especially NP-30 containing 7 mg/mL of propolis) can preserve oxidative balance. These preparations increased glutathione, catalase, and glutathione peroxidase activities, as well as Bcl-2 (anti-apoptotic B cell leukemia/lymphoma-2 protein), and decreased malondialdehyde levels and Bax (pro-apoptotic Bcl-2-associated X protein) in the testes of the treated rats by cisplatin. The sperm motility in the control, cisplatin, and cisplatin + NP-30 groups were 60%, 48.75%, and 78%, respectively. NP-30 administration completely corrected the deterioration in sperm features induced by cisplatin. These results demonstrated the significant benefits of the administration of the nanoform of propolis as co-therapy to alleviate the oxidative damage caused by cancer therapy. Moreover, the rats that were administered cisplatin and propolis nanoform had better indexes of the malondialdehyde level and glutathione peroxidase and catalase activities of testicular tissues. Such a positive effect could be explained by better solubility of the nanoform than propolis [39]. The ethanolic extract of propolis (1 to 10) increases activities of mitochondrial respiratory complexes II and IV of the human spermatozoa in vitro without an influence on mitochondrial membrane potential. Therefore, propolis has the ability to improve sperm motility [85]. These features of propolis could be used for the development of urological preparations for improving sperm motility and alleviation of oxidative damages caused by cancer therapy. Finally, the antioxidant activity of propolis is of interest to the pharmaceutical industry for the elaboration of different dosage forms for the prevention or treatment of diseases related to oxidative stress (diabetes, cancer, inflammation, cardiovascular diseases, and many others).

The addition of 0.06% propolis extract to chicken meat (breasts and thighs) confirmed that the quality of the meat was significantly higher than without this extract due to its oxidative stability, color, and sensory parameters. Moreover, the addition of natural antioxidants in the form of propolis extracts to chicken meat will enrich the food chain of humans with natural polyphenols [62].

### 3.2. Antimicrobial Activity 

#### 3.2.1. Antibacterial and Antifungal Activities

As antibacterial resistance is a serious modern threat to the health of people, beekeeping products as well as essential oils or other natural products are regarded as valuable sources of new antimicrobial medicinal agents [16,27,28,43].

Numerous studies confirm the antimicrobial activity of propolis of different types against a wide spectrum of microorganisms [16,26,43,44,66]. In fact, the antibacterial activity of propolis is connected with flavonoids and terpenes [26,37,66]. For example, flavon apigenin and sesquiterpene alcohol *tt*-farnesol can inhibit the plaque formation process. The mechanism of apigenin activity in preventing plaque formation is connected with the inhibition of glucosyltransferase enzyme activity in *Staphylococcus mutans*, which synthesizes glucans from dietary sucrose and, respectively, increases the pathogenic potential of dental plaque by promoting the adherence and accumulation of cariogenic *Streptococci* on the tooth surface. Moreover, sesquiterpene alcohol (3,7,11-trimethyl-2,6,10-dodecatrien-1-ol), *tt*-farnesol has high antibacterial capability to inhibit the growth and metabolism of *Staphylococcus mutans* by the disruption of bacterial membrane [88]. However, the other authors suppose that the antimicrobial activity of propolis is related to the synergism of all major and minor substances [26,66].

The Greek samples of propolis with the highest percentage of diterpenes (80.4%, 73.3%, and 81.9%) showed the highest antimicrobial activity against all the tested microorganisms, while they demonstrated specific strong activity against Gram-positive bacteria (*Staphylococcus aureus*, *Staphylococcus epidermidis*, and *Streptococcus mutans*) [37]. One more study showed that the propolis taken from the different parts of Turkey and Greece had effective antimicrobial activity against *Paenibacillus larvae*, a spore forming Gram-positive bacterium that infects honey bee colonies [26]. 

The red propolis showed MIC values in the range of 128 mg/mL to 512 mg/mL against *Escherichia coli* strains and 64 mg/mL to 1024 mg/mL against *Staphylococcus aureus* strains. However, the MIC values were all the same, 512 mg/mL, for the *P. aeruginosa* strains [14]. Neto et al. indicate such a tendency of red propolis: Gram-positive bacteria are more sensitive to propolis compared to Gram-negative bacteria, which is in agreement with the other studies [4,66,89,90]. All the tested *Candida* species were susceptible to lyophilized red propolis alcoholic extract, even resistant to fluconazole. Propolis was collected in Paraiba (Brazil’s northeast region) [89]. Hudz et al. showed that the Ukrainian propolis was not effective against *Escherichia coli* propolis [4]. Turkish propolis showed an inhibitory activity at higher concentrations against *Escherichia coli* and *Pseudomonas aeruginosa* compared to *Candida albicans* and *Staphylococcus aureus* [90]. The results obtained in the studies with Romanian poplar propolis showed a more noticeable antibacterial activity against Gram-positive bacteria compared to Gram-negative bacteria [66].

Aga et al., 1994 isolated three compounds with antimicrobial properties from Brazilian propolis. These compounds were identified as 3,5-diprenyl-4-hydroxycinnamic acid, 3-prenyl-4-dihdrocinnamoloxycinnamic acid, and 2,2-dimethyl-6-carboxyethenyl-2H-1-benzopyran. Their antimicrobial activity against *Bacillus cereus*, *Enterobacter erogenous*, and *Arthroderma benhamiae* was investigated. These authors found that the first compound demonstrated the highest activity. It is supposed that this first compound is one of the principal antimicrobial compounds in Brazilian propolis [87]. However, these authors did not indicate the type of Brazilian propolis (red, green, or brown). Fosquiera et al. revealed that Brazilian green and brown propolis has good antimicrobial activity against *Staphylococcus aureus*, *Candida albicans*, and *Pseudomonas aeruginosa*; 12% alcoholic and aqueous brown propolis solutions, 12% alcoholic green propolis solution, and 0.12% chlorhexidine were compared in term of their antimicrobial activity. The results of this study showed that the antimicrobial effect of propolis against *P. aeruginosa* and *S. aureus* was similar to chlorhexidine. However, green propolis was more effective than brown propolis. The antifungal effect was less. This study suggested that propolis may be regarded as an alternative medicinal product in controlling microorganisms, for instance, of the oral cavity [91]. Ghosh et al. revealed the complete inhibition of *Pseudogymnoascus destructans* spore germination by propolis [81].

The antimicrobial activity of propolis depends on many variables related to its origin such as the content of phenolics and flavonoids. The majority of the samples out of 24 from the different regions of Croatia showed potent antimicrobial activity against *S. aureus* and *C. albicans*. Gajger et al. demonstrated that the antimicrobial activity of the Croatian propolis is correlated with its components. The results showed that no sample exhibited activity against the Gram-negative microorganism *E. coli*, while 19 samples were effective against *Aspergillus niger*. These samples had MIC values of 6.25 and 12.5 mg/mL, respectively. The majority of samples had MIC values within the range (0.391–12.5 mg/mL) for Gram-positive bacterium *S. aureus* and yeast *Candida albicans* (0.391–12.5 mg/mL) that could be considered as the potential for therapeutic purposes. Gajger et al. revealed the significant correlation between the content of tectochrysin, galangin, pinocembrin chrysin, apigenin, and kaempferol in propolis and the antimicrobial activity against *Staphylococcus aureus* and the content of *p*-coumaric acid, apigenin, and kaempferol and the activity of propolis against *Candida albicans* [44]. 

Ota et al., 2001 studied the antifungal activity of propolis on 80 strains of *Candida* yeasts: 20 strains of *Candida albicans*, 20 strains of *Candida tropicalis*, 20 strains of *Candida krusei*, and 15 strains of *Candida guilliermondii*. Clear antifungal activity with the following order of sensitivity was revealed: *C. albicans* > *C. tropicalis* > *C. krusei* > *C. guilliermondii*. The patients with full dentures who used a hydroalcoholic propolis extract showed a decrease in the number of *Candida* [92].

Therefore, there is significant proof that propolis possesses antimicrobial activity that is a prerequisite for the development of new propolis preparations, which could be employed for controlling various diseases induced by multiresistant microorganisms, especially for the treatment of ailments of the oral cavity. Actually, all the types of propolis and individual components of propolis have been widely investigated in the studies directed on its administration in dentistry [65,88,93,94,95,96,97,98,99]. Propolis was assessed as an intracanal medicament, capping agent, antimicrobial and anti-biofilm agent against microbial species associated with caries, periodontal disease, and *Candida* infections [93,95]. Propolis ethanol extracts inhibit the growth of cariogenic bacteria, which include mainly *Staphylococcus mutant* and *Staphylococcus sobrinus* [96,97]. The dried ethanolic propolis extract has advantages over calcium hydroxide as a capping agent in vital pulp therapy. In addition, it induced the stimulation of stem cells and the production of high quality tubular dentin [95]. 

Propolis was comparable with triple antibiotic paste (a mixture of the contents of a doxycycline 100 mg capsule and ground metronidazole 500 mg and ciprofloxacin 250 mg tablets) in the disinfection treatment in the regenerative endodontic. Propolis as a root canal orifice plug induced a significant increase in root length and dentin thickness, and a diminishing in apical diameter similar to those of mineral trioxide aggregates after the revascularization of necrotic immature permanent teeth in dogs. One of the reasons for such an effect is the reduction in the number of microorganisms inside the root canal. The Egyptian propolis came from El Monofia province. The frozen propolis was ground and dissolved in 96% ethanol at a ratio of 1 to 1. The mixture was incubated for 2 weeks at a temperature of 30 °C and filtered. Then the filtered mixture was concentrated at a temperature of 30 °C for 6 h. The final extract of propolis had a density of 150 mg/mL. For the preparation of the propolis orifice plug, 1–2 drops of glycerin were added to the final extract of propolis (150 mg). The mixture was mixed until a thick paste, applicable for orifice plugging, was obtained [98].

Üstün et al. indicated that the combined use of AH Plus with propolis as an intracanal medicament was looking quite promising for favorable sealer–dentin interfacial bond strength. In their study, forty recently extracted human maxillary central incisors with completely formed apices and straight canals were used [83].

Toothpaste containing propolis reduced dental plaque scores, as there was a meaningful difference in the dental plaque score between the control and treatment groups. Lower dental plaque scores were observed in patients who brushed their teeth with this toothpaste compared to those in the control group. The lower dental plaque scores can indicate the inhibition of dental plaque formation [99].

Forty chronic periodontitis patients were randomly divided into two groups for the treatment. The sites were treated by scaling and root planing followed by gingival irrigation with physiological saline in the control group (*n* = 20). In the test group (*n* = 20), the sites were treated by scaling and root planing followed by the subgingival placement of ethanolic propolis extract. The results of the study showed significant bactericidal activity of the Spanish propolis extract on *Porphyromonas gingivalis* and *Tannerella forsythensis* in bacterial counts one month after periodontal therapy. Overall, these results suggest prophylactic and therapeutic administration of ethanolic propolis extract against periodontal diseases, improving clinical parameters, reducing gingival bleeding, and decreasing counts of periodontopathogenic bacteria. The subgingival administration of ethanolic propolis extract represents a promising modality as an adjuvant in periodontal therapy to avoid microbial resistance and other adverse effects [65]. The antimicrobial and anti-biofilm activities indicated the potential of propolis as the component in oral health care products.

#### 3.2.2. Antiviral Activity

Various studies demonstrate that propolis extract exerts antiviral activity against a diverse range of DNA and RNA viruses such as herpes simplex virus type 1, herpes simplex virus type 2 (HSV-2), adenovirus type 2, vesicular stomatitis virus, and poliovirus type 2 [100,101] that can be the base for the development of tablets, sprays, and drops for the oral cavity. Propolis flavonoids (quercetin, kaemferol, and chrysin) reduced the replication and even infectivity of some strains of herpes virus, adenovirus, rotavirus, and coronavirus (OC43) [88]. Moreover, quercetin with vitamin C as aminopeptidase inhibitors are considered as the active substances for the treatment of SARS [102]. In addition, the in vitro activity against herpes simplex virus type 1 of 3-methyl-but-2-enyl caffeate isolated from poplar buds or prepared by synthesis decreased the viral titer by 3 log10, and viral DNA synthesis by 32-fold; 3-methyl-but-2-enyl caffeate is a component of propolis [103].

It was revealed that mixtures of different volatiles revealed a much higher selectivity index as the alternative antiviral agents than their isolated components [104].

Kumar’s study based on molecular docking revealed that caffeic acid phenethyl ester (CAPE) (one of the bioactive propolis ingredients) possessed the potential to inhibit the functional activity of SARS-CoV-2 protease (a very important protein for the virus survival). CAPE was interacting with the highly conserved residues of the proteases of coronaviruses [105]. Although there is a crucial need for experimental and clinical studies, the authors deem that their results showed extraordinary therapeutic value for the management of COVID-19 [92]. Therefore, standardized propolis extracts could be used for the complementary or prevention treatment of patients with SARS CoV-2.

The large potential of hydromethanolic extract of geopropolis from *Scaptotrigona postica* in reducing the copies of the DNA of herpes simplex virus type 1 (HSV-1) at low concentrations can be explained by the known antiviral activity of *C*-glycosylflavones, catechin-3-*O*-gallate, and 3,4-dicaffeoylquinic acid. The hydromethanolic extract of geopropolis inhibited the viral replication and the entry of the virus into cells [61].

It was revealed that a higher efficacy of the ethanolic extract of Mexican propolis against canine distemper virus was achieved when it was administered before the viral infection. This indicated that the extract directly interacted with host cells by interfering with proper recognition between cellular receptors and virus proteins. Therefore, the extract prevented virus internalization and further replication. Quercetin administered at the same time of infection increased cell viability and diminished viral gene expression. This flavonoid is the subject of many studies as it is generally revealed in all types of propolis. Considering the fact that quercetin has a higher antiviral effect if it is administered simultaneously with the infection, it is supposed that it inhibits the intracellular phase of the replication cycle of canine distemper virus and viral polymerase and interferes with viral nucleic acid synthesis [38].

### 3.3. Anti-Inflammatory Activity

Inflammation is a human host-defense mechanism against internal or external danger signs, which can be activated by different stimuli. Among such stimuli are physical injuries and bacterial products. Conventional inflammatory cytokines mediate inflammation in order to eliminate the invading microorganisms or damaged cells, promote tissue repair and regeneration. However, uncontrolled inflammation can cause massive macrophage activation resulting in self-inflicted death, which subsequently triggers extensive neutrophil recruitment, thereby inducing severe immunopathologies [106].

Such identified extracted flavonoids from Nepalese propolis as 3′,4′-dihydroxy-4-methoxydalbergione, 4-methoxydalbergion, cearoin, and chrysin significantly inhibited the IL-33-induced mRNA expression of inflammatory genes, including IL-6, tumor necrosis factor-alpha (TNF-α), and IL-13 in bone marrow-derived mast cells. The IL-33-induced activation of nuclear factor κB (NF-κB) was also inhibited by these four flavonoids, which was consistent with their inhibitory effects on cytokine expression [82].

Neovestitol is one of the principal bioactive components of Brazilian red propolis, which has anti-inflammatory, antimicrobial, and antioxidant effects. RAW264.7 murine macrophages activated with LPS were treated with neovestitol. Neovestitol at 0.22 mM inhibited NO production by 60% without affecting cell viability and diminished the levels of pro-inflammatory mediators such as granulocyte-macrophage colony-stimulating factor (GM-CSF), IFN-γ, IL-1β, IL-4, TNF-α, and IL-6, whereas it increased IL-10 production. These cytokine profile changes were associated with the downregulation of transcription of genes involved in nitric oxide production, NF-kB, IL-1β, and TNF-α signaling pathways [106].

The subcutaneous administration of propolis flavonoid liposomes with ovalbumin to mice effectively activated the cellular and humoral immune response, including causing higher level concentrations of IgG, IL-4, and interferon-gamma (IFN-α) in serum and the proliferation rates of splenic lymphocytes. After the administration of propolis flavonoid liposomes, significantly greater concentrations of IL-1β, IL-6, and IFN-γ were observed compared to only propolis flavonoids or liposomes. IL-1β, IL-6, and IFN-γ can facilitate different immune responses, including strikingly enhancing antigen-driven responses of CD4 and CD8 T cells [107]. 

Kurek-Górecka et al. [10] supposed that curative properties of propolis and other beekeeping products which could be applied to the skin are due to the content of flavonoids and other phenolic compounds. Its anti-inflammatory, disinfectant, and antiviral properties are very useful for treating burn wounds. Burn treatment with the propolis ointment led to enhanced collagens and its components’ expression in burn wounds of the inflicted pigs [10,108]. Propolis has anti-aging properties due to its ability to smooth out wrinkles [10,109].

Shinmei et al., 2009 studied the effect of Brazilian propolis after the oral administration on sneezing and nasal rubbing in the experimental allergic rhinitis of mice. It was concluded that propolis may be effective in the relief of the symptoms of allergic rhinitis through inhibition of histamine release from rat mast cells induced by both antigens. A single administration of propolis caused no significant effect on both antigen-induced nasal rubbing and sneezing at a dose of 1000 mg/kg, but a considerable inhibition was observed after the repeated administration of this dose for two weeks [110]. This activity of propolis is important, taking into consideration that rhinitis is a global health problem, affecting social life, sleep, school, and work performance, regardless of gender, age, and ethnic background [111].

A principal role in the pathogenesis of bronchial asthma belongs to oxidative stress. Reactive oxygen species perform an important function in the pathogenesis of airway inflammation. After an inflammatory stimulus, NF-κB in the nucleus induces the expression of a wide variety of genes in inflammation, including cytokines (e.g., IL-1, IL-6, and TNF-α), enzymes (including nitric oxide synthase), adhesion molecules, and acute-phase proteins. CAPE modulates the intracellular generation of ROS and redox-sensitive transcription factor NF-κB, namely administration of CAPE showed an inhibitory effect on the generation of ROS. CAPE (10 mg/kg/day) weakened the allergic airway inflammation and hyperresponsiveness in a murine model of ovalbumin-induced asthma. Therefore, CAPE could be useful for the adjuvant therapy of bronchial asthma [112]. Khayyal et al. (2003) studied the effect of an aqueous extract of propolis (one time per day for 2 months) to patients with mild to moderate asthma. As a result, the incidence and severity of nocturnal attacks were reduced, and ventilatory functions were improved in the propolis-treated patients, which was related to the decreased levels of prostaglandins, leukotriene, and proinflammatory cytokines (TNF-α, IL-6, and IL-8) and increased levels of ꞌꞌprotectiveꞌꞌ cytokine IL-10. The patients on the placebo preparation did not show significant improvement in the ventilatory functions or in the levels of mediators. Khayyal et al. explained this effect by the presence of CAPE in propolis [113]. The advantage of such a preparation was its standardization for the content of organic aromatic acids, mainly caffeic, ferulic, iso-ferulic, cinnamic, and 3,4-dimethoxy-cinnamic acids (not less 0.05%) in addition to trace amounts of various flavonoids. However, this preparation was made from propolis collected in different countries, such as Denmark, China, Uruguay, and Brazil. In addition, Borrelli et al. related the anti-inflammatory activity of the ethanolic propolis extract to CAPE in the carrageenin edema [114].

Therefore, propolis preparations and/or some isolated compounds from propolis can be regarded as potential candidates to modulate chronic inflammation in humans.

### 3.4. Anticancer Activity

Propolis can be used for anticancer adjuvant therapy due to its cytotoxic effects [115,116,117,118]. 2′-Hydroxy-3′,4′-dimethoxychalcone as a component of propolis was reported to show, in vitro, 100% and 97% cytotoxic effects to Ehrlich ascite cancer cells and Dalton’s lymphoma ascite cells, respectively. 2′,4′-Dihydroxy-3′-methoxychalcone showed very strong cytotoxicity, IC_50_ values < 4 µg/mL, against P-388 and HT-29 cancer cell lines, from strong to moderate cytotoxic activity towards HeLa, HL-60, and MCF-7 cancer cell lines with IC_50_ values of 12.2, 5.1, and 12.5 µg/mL, respectively [15]. 

Such prenylflavanones as propolin A and B isolated from Taiwanese propolis induced apoptosis human melanoma, C6 glioma, and HL-60 cells [53]. Ishida et al. indicate that CAPE is a very important component of propolis with anticancer activity in vivo. Propolis extracts with moderate levels of CAPE (~5%) could be used for cancer treatment in appropriate experimental models. However, CAPE is easily degradable. Thermostable complexes of CAPE and propolis with γ-cyclodextrin (γ-CD) are better for the studies in the treatment of cancer and other ailments [116]. It seems that the standardization of propolis preparations for CAPE content is very important for the development of medicinal products of propolis for anticancer therapy.

The treatment with ethanol-extracted Chinese propolis (EECP) in the different concentrations of 25, 50, and 100 μg/mL, and CAPE (25 μg/mL), significantly inhibited LPS-stimulated MDA-MB-231 cell line proliferation, migration, and NO production. Moreover, EECP and CAPE activated caspase 3 and PARP to induce cell apoptosis and also upregulated LC3-II and decreased p62 levels to induce autophagy during the process. In general, EECP and its major constituent, CAPE, inhibited breast cancer MDA-MB-231 cell proliferation in the inflammatory microenvironment through the activation of apoptosis and autophagy and by inhibiting the TLR4 signaling pathway. Therefore, standardized propolis extracts and CAPE may be preparations in treating inflammation-induced tumors [117]. 

The extracts of poplar propolis from three continents, including Europe and Asia, were mildly cytotoxic toward cancer cells, in particular osteosarcoma cells (IC_50_: 81.9–86.7 µg/mL). Despite this, cytotoxicity was observed also in non-tumor L929 cells, with lower values of IC_50_. IC_50_ values were, respectively, 126.0, 185.8, and 149.6 μg/mL. However, the other normal human mesenchymal cells (hMSCs) demonstrated the lowest sensitivity to propolis (IC_50_: 258.3–287.2 µg/mL). Moreover, the calculated IC_50_ values for three extracts were relatively close, respectively, namely 263.2, 258.3, and 287.2 μg/mL. The cytotoxicity of the propolis extracts on human monocytic leukemia (THP-1) cells was slightly stronger compared to that observed in the case of hMSCs, and the IC_50_ values measured for three extracts were, respectively, 187.4, 203.2, and 164.4 μg/mL. Leukemic HL60 cells showed a susceptibility similar or even greater than THP-1 cells to the propolis extracts. In THP-1 cells, extracts stimulated apoptosis caspase 3/7 activity. It is worth noting that the authors of this study accented that the reliable control over propolis composition from batch to batch was of supreme importance, and the standardization of the extraction conditions appears crucial [118]. 

Some described preparations of propolis with the indication of their activity are presented in Table 2.

We provide the discussed biological activities of different types of propolis in Figure 5.

## 4. Conclusions

The chemical compositions of propolis point out the fact that it is a potential source of natural compounds with many pharmacological effects and therapeutic applications. Propolis possesses the broad range of activities, including antibacterial, antifungal, antiprotozoal, hepatoprotective, antioxidant, anti-inflammatory, antiviral, anticancer, etc., properties. At present, it could be more widely used as a starting raw material for manufacturing extracts as active pharmaceutical ingredients and, thus, pharmaceutical products, including antimicrobial medicinal products. Choosing suitable solvents and conditions for solubilization of propolis of appropriate origin, it is possible to prepare standardized food or pharmaceutical products. In addition, the identification and determination of the compounds responsible for the biological activity of propolis of appropriate origin will facilitate the development of standardized preparations, thus ensuring their higher quality and efficacy. Moreover, propolis could be regarded as a source for isolation of biologically active substances. The current literature review also suggests that propolis may be used for its potential antimicrobial, anti-inflammatory, and anticancer properties in such dosage forms as mouthwashes, capsules, and tablets for the oral cavity and oral administration, a root canal orifice plug, and a liposome for parenteral administration. Finally, this study could be of interest for the food industry for the development of nutritionally well-balanced products. The results of this review paper indicate that propolis deserves to be better studied for its promising therapeutic effects from the point of view of the locality and its vegetation, standardization, nonclinical studies, and clinical trials. Moreover, regulatory agencies should establish quality parameters for propolis of a certain country. Furthermore, this review can partly overcome the problem of the standardization of propolis preparations, which is a main obstacle to wider use of propolis in the pharmaceutical industry.

## Figures and Tables

**Figure 1 molecules-27-01600-f001:**
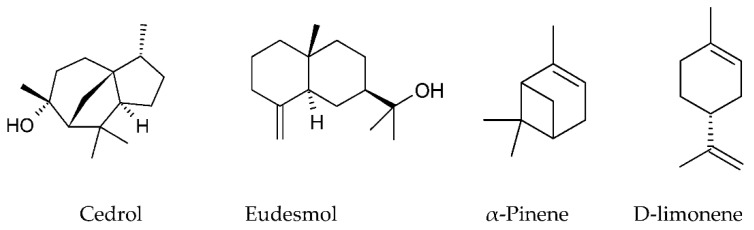
The structures of some mono- and sesquiterpenoids.

**Figure 2 molecules-27-01600-f002:**
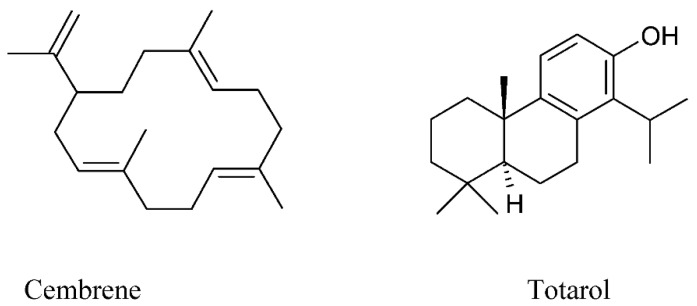
The structures of some diterpenoids.

**Figure 3 molecules-27-01600-f003:**
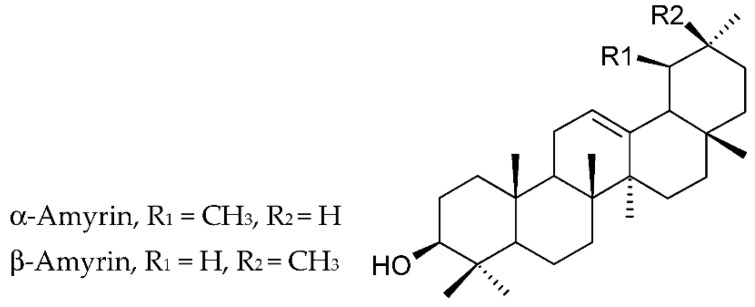
The structures of amyrins.

**Figure 4 molecules-27-01600-f004:**
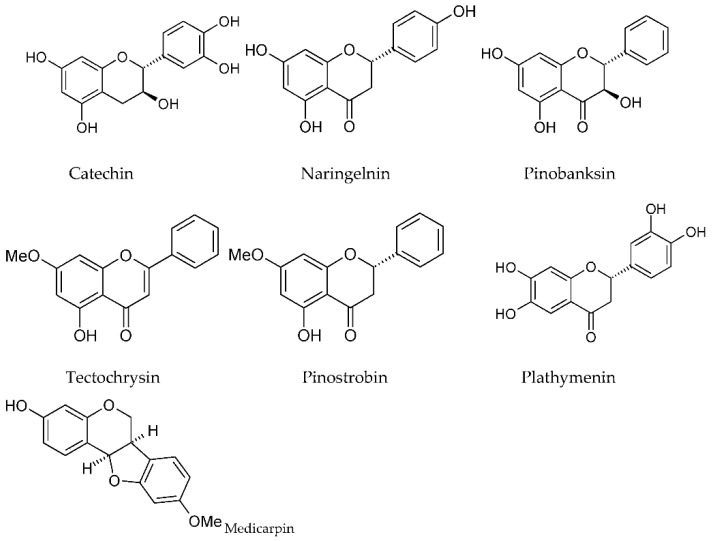
The structures of some flavonoids identified in propolis.

**Figure 5 molecules-27-01600-f005:**
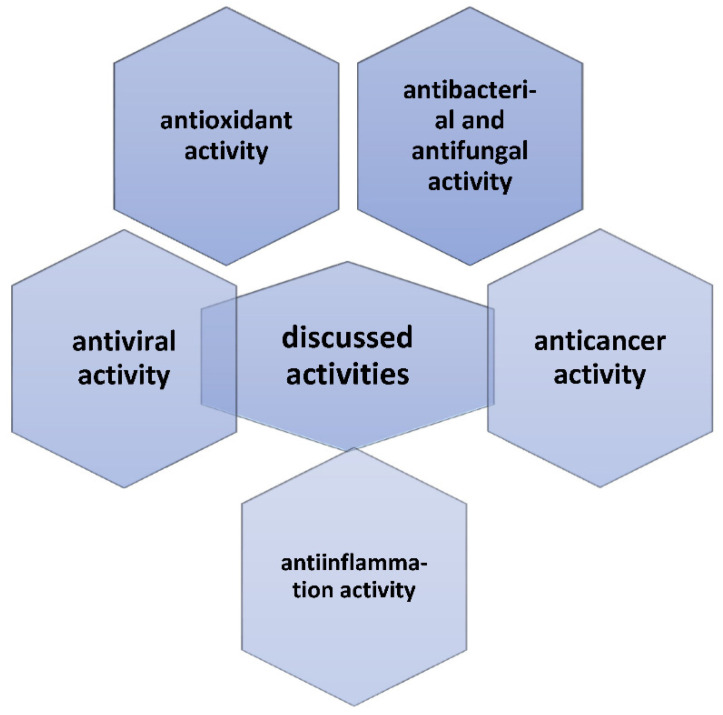
The biological activities discussed in this paper.

**Table 1 molecules-27-01600-t001:** The main biologically active compounds found in the different propolis samples.

Group	Representatives	Propolis Type	Reference
Terpenoids			
Mono- and sesquiterpenoids	Cedrol (17.0%) α-Pinene (56.1%)	Algerian, Oum El Bouaghi Batna	[67]
	α-Pinene and β-pinene	Brazilian	[18]
	Limonene	Venezuelan	[68]
	β-Caryophyllene and nerolidol	Argentinian green	[68]
	Cedrol, γ-eudesmol, phenethyl alcohol, benzyl alcohol, 2-3,4-dimethoxystyrene, methoxy-4-vinylphenol, and guaiol	Chinese	[69]
Diterpenes	Clerodane diterpenes	Brazilian	[75]
	Cembrene (C_20_H_32_) and totarol (C_20_H_30_O)	Saudi Arabia (Baha)	[30]
Triterpenoids	α-Amyrin	Ethiopian propolis	[64]
	Lupenone, α-amyrin and β-amyrin	Cameroonian	[76]
Fatty acids			
Unsaturated fatty acids	*cis*-11-Eicosenoic acid (C20:1, *n*-9), *cis*-11,14-eicosadienoic acid (C20:2, *n*-6), *cis*-5,8,11,14,17-eicosapentaenoic acid (C20:5), arachidonic acid (C20:4, *n*-6), linoleic acid (C18:2, c + t *n*-6), palmitoleic acid (C16:1), and γ-linolenic acid (C18:3, *n*-6)	Algerian	[21]
Saturated fatty acids	Oleic acid, nonanoic acid (C_9_H_18_O_2_), decanoic acid (C_10_H_20_O_2_), dodecanoic acid (C_12_H_24_O_2_)	Saudi Arabia	[30]
	Hexatriacontanoic acid (C_36_H_72_O_2_)	Cameroonian	[76]
Flavonoids			
Flavanols	Catechin	Mexican	[38]
Flavanones	Pinocembrin	Mexican	[38]
		European brown poplar	[42]
		Turkish (1.22 mg/g)	[39]
		Croatian (0–6.39 mg/mL)	[44]
		Irish, Czech, German	[43]
		Romanian poplar	[66]
		Polish	[63]
	Naringenin	Mexican	[38]
	Pinostrobin	Turkish (2.93 mg/g)	[39]
	Plathymenin	Nepalese	[82]
Flavanonols	Pinobanksin 3-acetate	Australian	[15]
	Pinobanksin-3-*O*-propionate, pinobanksin-3-*O*-butyrate, pinobanksyn-3-*O*-pentenoate	European brown poplar	[42]
	Pinobanksin	Turkish	[39]
		Polish	[63]
Flavones	Chrysin	Polish	[19,63]
		Mexican	[38]
		European brown poplar	[42]
		Turkish (2.94 mg/g)	[39]
		Croatian (0–8.02 mg/mL)	[44]
		Irish, Czech	[43]
		Romanian poplar	[66]
	Apigenin	Brazilian red	[14]
		European brown poplar	[42]
		Croatian (0–1.23 mg/mL)	[44]
		Polish	[63]
	Luteolin	Turkish	[12]
		Brazilian red	[14]
	Tectochrysyn	Croatian (0–16.07 mg/mL)	[44]
*C*-glycosyl flavones	Apigenin-6,8-di-*C*-malonyl glucoside dihexoside isomer, apigenin-di-*C*-malonyl trihexoside isomer	Geopropolis from *Scaptotrigona postica*	[61]
Flavonols	Quercetin	Brazilian red	[14]
		Mexican	[38]
		European brown poplar	[42]
		Romanian poplar	[66]
		Polish	[63]
	Kaempferol	Mexican	[38]
		European brown poplar	[42]
		Croatian (0–0.672 mg/mL)	[44]
		Polish	[63]
	Galangin	Polish	[19,63]
		Turkish (0.09 mg/g)	[39]
		Croatian (0–8.71 mg/mL)	[44]
		Irish, Czech	[43]
		Romanian poplar	[66]
	Rutin	Turkish	[12]
		Brazilian red	[14]
Isoflavones	Homopterocarpin, medicarpin, 4,7-dimethoxy-2-isoflavonol	Brazilian red	[34]
	Medicarpin	Nepalese	[82]
Chalcons	2′,3′,4′-Trimethoxychalcone, 2′-hydroxy-3′,4′-dimethoxychal-cone, 2′,4′-dihydroxy-3′-methoxychalcone	Australian	[15]
Phenolic acids	Caffeic acid	Turkish (0.17 mg/g)	[12,39]
		Algerian	[21]
		Polish	[63]
	Ferulic acid	Turkish (0.36 mg/g)	[12,39]
		Croatian (0–1.370 mg/mL)	[44]
		Romanian poplar	[66]
		Polish	[63]
	*t*-Cinnamic acid	Turkish (0.05–0.14 mg/g or 3.95 mg/g)	[12,39]
	*p*-Coumaric acid	Croatian (0–1.031 mg/mL)	[44]
		Romanian poplar	[66]
		Polish	[19,63]
		Algerian	[21]
		Turkish (0.07–0.24 mg/g)	[12]
	Chlorogenic acid	Algerian	[21]
	Gallic acid	Algerian	[21]
		Turkish (0.015–0.025 mg/g)	[12]
	Syringic acid	Turkish	[12]
		Polish	[63]
	Vanilic acid	Polish	[63]
	3,5-Diprenyl-4-hydroxycinnamic acid, 3-prenyl-4-dihydrocinnamoloxy-cinnamic acid (C_23_H_24_O_4_)	Brazilian propolis	[87]

**Table 2 molecules-27-01600-t002:** Some pharmaceutical products tested in different studies.

Product	Composition and Brief Technology	Activity	Reference
Toothpaste with propolis	There is no information in the paper	Reducing dental plaque formation	[105]
Propolis extract (Spanish propolis)	The propolis extract was prepared under aseptic conditions; 20 g of unrefined propolis was crushed and dissolved in 100 mL of 66% ethanol. The mixture was kept at room temperature for 28 days and subsequently filtered	Antibacterial activity against anaerobic bacteria (*Porphyromonas gingivalis* and *Tannerella forsythensis*) using sublingual administration	[65]
Nanoform of Turkish propolis	A total of 3.5 g of chitosan was dissolved in 230 mL of 2% aqueous acetic acid solution (*v*/*v*) in an ultrasonic bath; 1 g of tween 80 was added into the chitosan solution which was mixed by magnetic stirring at a temperature of 25 °C for 30 min. Then, 840 mg of propolis was dissolved in 120 mL of ethanol. This propolis solution was added into the chitosan/tween 80 blend, and sonicated for 10 min in order obtain NP-10	Propolis-bearing polymeric nanoparticles can mitigate the side effects of cisplatin	[39]
Liposoms for subcutaneous administration	Liposoms contains a complex mixture which principally contains rutin, myricetin, quercetin, kaempferol, apigenin, pinocembrin, chrysin, and galanigin. Propolis was extracted with 95% ethanol three times, and the ethanol solution was retrieved. Then, the precipitation was extracted with ethyl acetate three times, and ethyl acetate was retrieved. At the end, the precipitation was dried in vacuum, and propolis flavonoids were obtained	Immunological enhancement activity	[107]
Propolis apitherapeutic ointment	There is no information in the paper	Propolis burn treatment led to enhanced collagens and its components expression	[108]
Propolis granular (Yamada Apiculture Center, Inc., Okayama, Japan) and propolis ethanol extract 55 wt.%/vol.% (for oral administration)	Propolis granular was dissolved in 5% gum arabic, and propolis ethanol extract was dissolved in 1% ethanol	The relief of symptoms of allergic rhinitis through inhibition of histamine release	[110]
Dried 13% solution of the aqueous extract of propolis	A 13% solution of the aqueous extract of propolis was supplied by Propharma (Stenlose, Denmark), which was prepared by aqueous decoction of crude propolis, collected from Denmark, China, Uruguay, and Brazil. This extract was standardized to contain not less than 0.05% of organic aromatic acids, chiefly caffeic, ferulic, isoferulic, cinnamic, and 3,4-dimethoxy-cinnamic acids in addition to trace amounts of various flavonoids. The aqueous extract was first concentrated, then spray-dried under high pressure before being incorporated into the milk formula. The sachets were intended to be given suspended in water as a milk drink orally once a day for two months.	Marked reduction in the incidence and severity of nocturnal attacks and improvement of ventilatory functions	[113]
propolis-γ CD powder	The technology is not described	Anticancer	[116]

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
