# Peer review of "Chemical Variability and Pharmacological Potential of Propolis as a Source for the Development of New Pharmaceutical Products"

_molecules, 2022, doi:10.3390/molecules27051600_

Round 1

Reviewer 1 Report

In attention of the journal Authors,

In this review (molecules-1590250), the authors made substantial research efforts to survey the literature and summarize the latest updates on propolis, with a special focus on its potential as a raw material for the development and manufacture of dietary supplements as health-promoting products and different dosage forms.

The outcomes provided by the manuscript could be of real interest for the food industry dealing with the development of nutritionally well-balanced products and also highlight the opportunity of new investigations of propolis considering its promising therapeutic effects from a point of view of the locality and its vegetation, standardization, nonclinical studies, and clinical trials. The surveyed literature and protocol applied to achieve its purpose are adequate, well-structured, and well-presented.

Considering the potential impact of the manuscript outcomes in the research world, I recommend that the manuscript be accepted for publication in Molecules journal following minor revision.

Recommendations:

  1. Did the authors check in the literature if new propolis constituents were discovered or discussed after 2018? The authors have verified, synthesized, and discussed this aspect until 2018. Even if no constituents were discussed after 2018; this absence from 2018-2022 should be emphasized in some way.
  2. For text consistency, please choose whether to write (1) the words in uppercase or lowercase (Table 1, e.g. the flavonoids pinocembrin, Naringenin, Chrysin, etc), (2) delete dots and commas after references of Table 1, where appropriate (e.g. Saudi Arabia [30]. or Mexican [38],), (3) deliminate bold type from the reference of section 2.2, remove the additional space from references inserted in the main text (e.g [36, 38,83], [21, 30, 76] in section 2.5.2, [115, 116, 117, 118]. Etc section 3.5), (4) use only one version “IC50” or “IC50”, Please verified the entire text and correct it accordingly.
  3. Please correct the word “sourse” with “source” and “propois” with “propolis” and “Alsthough” with “Although”, delete “for”, use the same character's type, delete second “can”, in the sentences: “Therefore, propolis could be regarded as a sourse of omega ……” and “Some described preparations of propois are ……..”, “Alsthough there is a crucial need for experimental and clinical studies…..”, “Kumar´s study based on for molecular docking”, “…medicament was looking quite promising for favorable sealer-dentin”, “Khayyal explained this effect by the presence of CAPE….”, “….this review can partly can overcome the problem of the standardization of…” Please verified the entire text and correct it accordingly.
  1. The list of references, the journals names are written both abbreviated and extended. Please keep only one way of writing, the version indicated in the journal guide.

Author Response

Dear reviewer, thank you for your valuable remarks. All the explanations are in the attached file

Reviewer 2 Report

The review  "Chemical variability and pharmacological potential of propolis as a source for the development of new pharmaceutical products" has scientific merits but many points required more attention by the authors.

among these points:

1- The title of the review should be revised for typos and the whole manuscript should be checked by an English native speaker to improve the quality of the language.

2- The abstract should be rewritten to be more specific and give precise aim(s) of the review the methods used for selecting and collecting data and also report strong the most relevant key findings 

3- The introduction part should include parts about the most common methods used in extraction and the pharmaceutical dosage forms already containing propolis to match with the title and the subsequent contents.

4- In table 1: it is better to separate the last column into 2 columns one for the geographical source and one for the references. In the first section; mono- sesquiterpenes; the name of the compounds should be only given and the geographical source should be moved to its location

5- It is highly recommended to draw some representatives structures to help the audience in better understanding 

6- The biological activities should be rearranged so that closely related activities must be related together and a table or a figure summarizing the most key findings will be excellent 

Author Response

Dear reviewer, we are thankful for your remarks and recommendations . All the explanations are provided in the attached file

Round 2

Reviewer 2 Report

The manuscript "Chemical variability and pharmacological potential of propolis as a sourse for the development of new pharmaceutical products" has much improved and considerable effort has been done

only the structures should be revised and drawn by a the same software with the same style 

Author Response

Dear reviewer, we are very thankful for your review . We have done the structures with a special program. Also, we revised our manuscript. We hope that we improved our manuscript due to your valuable recommendations and remarks